# Cryo-EM structure of the *Saccharomyces cerevisiae* Rpd3L histone deacetylase complex

Avinash B. Patel[1] ✉, Jinkang Qing[1,2], Kelly H. Tam[1], Sara Zaman[1], Maria Luiso[1], Ishwar Radhakrishnan [1] ✉ & Yuan He [1] ✉

The Rpd3L histone deacetylase (HDAC) complex is an ancient 12-subunit complex conserved in a broad range of eukaryotes that performs localized deacetylation at or near sites of recruitment by DNA-bound factors. Here we describe the cryo-EM structure of this prototypical HDAC complex that is characterized by as many as seven subunits performing scaffolding roles for the tight integration of the only catalytic subunit, Rpd3. The principal scaffolding protein, Sin3, along with Rpd3 and the histone chaperone, Ume1, are present in two copies, with each copy organized into separate lobes of an asymmetric dimeric molecular assembly. The active site of one Rpd3 is completely occluded by a leucine side chain of Rxt2, while the tips of the two lobes and the more peripherally associated subunits exhibit varying levels of flexibility and positional disorder. The structure reveals unexpected structural homology/analogy between unrelated subunits in the fungal and mammalian complexes and provides a foundation for deeper interrogations of structure, biology, and mechanism of these complexes, as well as for the discovery of HDAC complex-specific inhibitors.

Lysine acetylation is a well-established post-translational modification that can profoundly alter the structure, function, and dynamics of proteins[1,2]. Acetylation levels are highly abundant in euchromatin, especially within the tail regions of core histones, and are positively correlated with enhanced DNA accessibility[3,4]. Steady-state levels of histone acetylation are determined by two enzyme families that add or remove this modification: histone acetyltransferases (HATs) and histone deacetylases (HDACs). These enzymes exist in giant multi-protein complexes and their actions are tightly regulated by the associated subunits[1,2]. Despite significant progress in understanding individual HDAC structure and function, including the development of novel inhibitors[5], no high-resolution structure of these enzymes in their native, fully assembled complexes has yet been described. Here we describe a near-atomic resolution structure of a prototypical HDAC complex found in budding yeast that has counterparts in other fungi, plants, and animals[6–10].

The gene for Rpd3 in budding yeast was the first HDAC ever cloned and subsequently shown to possess deacetylase activity like its mammalian homologues[11–13]. Rpd3 was later found to be a member of at least two major HDAC complexes, including the 1-2 MDa 12-subunit Rpd3L complex[6], implicated in promoter-specific localized deacetylation and transcription repression[14,15], and the 0.5-0.6 MDa five-subunit Rpd3S complex that functions in deacetylation and repression of transcription initiation from cryptic sites within intragenic regions[16–18].

Here we present the structure of the *S. cerevisiae* Rpd3L complex determined by cryogenic electron microscopy (cryo-EM). The complex contains an asymmetric dimeric core that can be divided into two lobes. Each lobe contains the subunits Sin3 and Rpd3 along with the flexibly tethered Ume1. The Dep1, Sds3, Sap30, Rxt2, Pho23, and Rxt3 subunits, along with Sin3, form the underlying scaffold for the complex. The active site of the sole catalytic subunit in the complex, Rpd3, is readily accessible in one lobe but is occluded in the other by a

[1]Department of Molecular Biosciences, Northwestern University, Evanston, IL, USA. [2]Interdisciplinary Biological Sciences Program, Northwestern University, Evanston, IL, USA. ✉e-mail: avinash.patel@northwestern.edu; i-radhakrishnan@northwestern.edu; yuanhe@northwestern.edu

conserved segment within the Rxt2 subunit. Finally, the structure also reveals that the chromatin- and repressor-binding domains found within various subunits are flexibly tethered to the structured core of the complex.

## Results & discussion

We purified the *Saccharomyces cerevisiae* Rpd3L complex using tandem affinity purification (TAP) approaches by taking advantage of the TAP tag on the Rxt2 subunit (Supplementary Fig. 1A). The purified complex was characterized by mass spectrometry for the presence of all 12 subunits (Supplementary Fig. 1B) and confirmed by HDAC assays to be active and responsive to known inhibitors such as suberoylanilide hydroxamic acid (SAHA; Supplementary Fig. 1C & 1D). The purified complex was subsequently characterized by negative-stain EM. 2D class averages of particles picked from electron micrographs revealed a structured core that was confirmed via 3D reconstructions (Supplementary Fig. 1E & 1F). Cryo-EM data were then acquired, processed, and the resulting map iteratively improved to 3.5 Å resolution (Supplementary Fig. 2). Following map validation, atomic models for the various subunits were built de novo or in conjunction with the models predicted by AlphaFold2 (Supplementary Figs. 3–5). The final refined model of the complex was generally in good agreement with

experimental data while maintaining close to ideal values for stereochemistry.

## Overall architecture of the Rpd3L complex and broad roles of individual subunits

The cryo-EM structure of the Rpd3L complex reveals an overall architecture comprising two giant lobes attached to the stems of a two-stemmed cross-brace scaffold (Fig. 1A, 1B). Lobe I is formed largely by Rpd3, Sin3, and Sap30 and is wedged between the stems of the cross-brace (Fig. 1C). Lobe II is made up of Rxt3 and a second copy of Rpd3 and Sin3, but unlike Lobe I, engages only one of the stems. Both Rpd3 copies are located near the base of each lobe and engage in extensive interactions with the stems (Fig. 1C). Stem I is made up of a heterodimeric, anti-parallel, two-stranded α-helical coiled-coil comprising Sds3 and Dep1. Stem II is comparatively shorter, formed by a heterodimeric, anti-parallel, four-stranded α-helical coiled-coil comprising Rxt2 and Pho23 oriented almost perpendicular to Stem I, completing the brace (Fig. 1C).

Each lobe and stem together appear to form an independent structural unit (Fig. 2A). This bipartite organization takes the form of an asymmetric dimer characterized by Sin3 and Rpd3 engaging with each other extensively and almost identically in each unit of the dimer

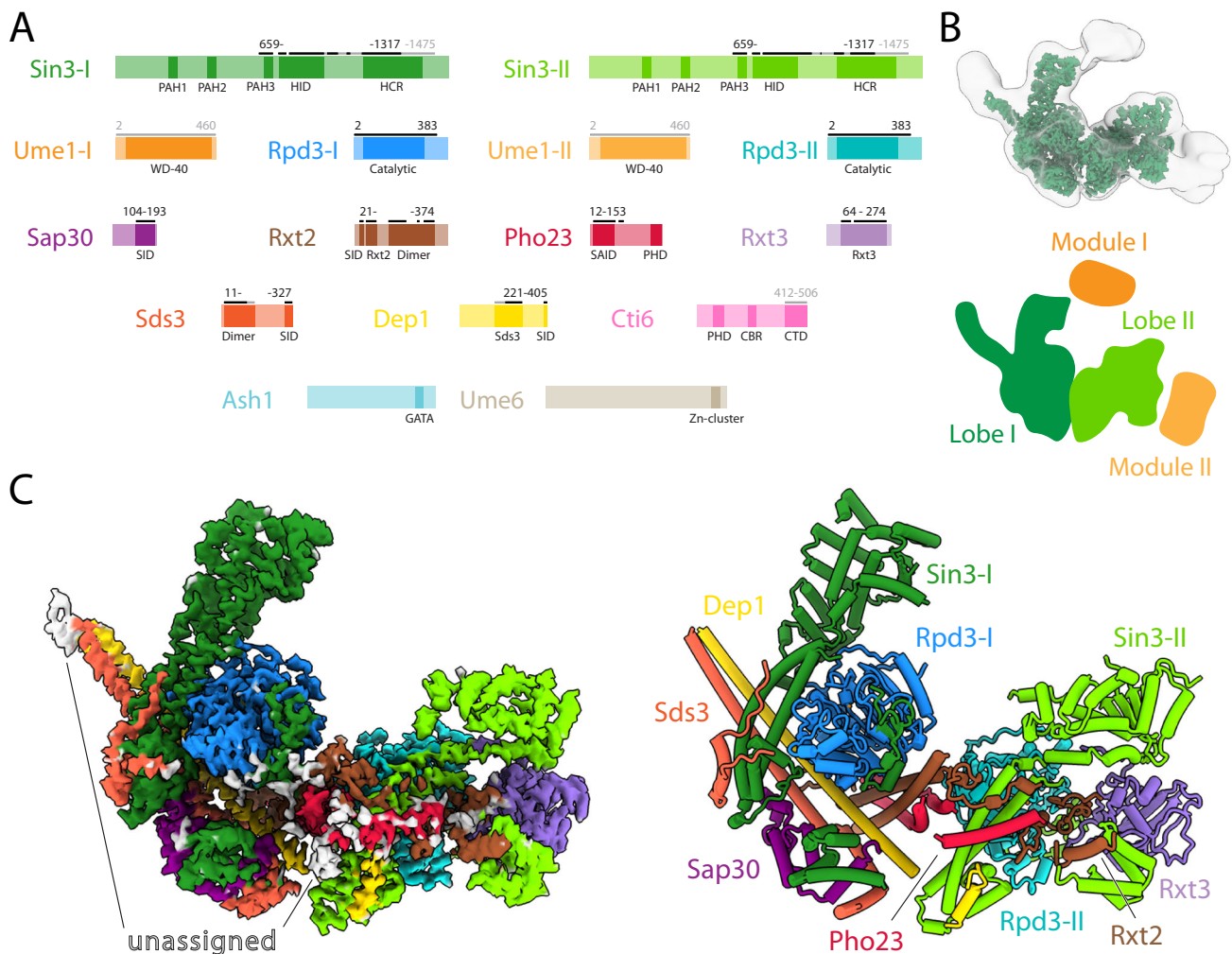

**Fig. 1 | Architecture of the Rpd3L complex. A** Domain maps of Rpd3L subunits. Modeled regions are marked with black bars, whereas extended regions based on AlphaFold2 predictions are marked with grey bars; numbers indicate starting and ending residues. **B** *Top*: Cryo-electron microscopy (cryo-EM) map in green showing the best-defined parts of Rpd3L. A transparent lower-threshold cryo-EM map is

overlaid to show the flexible density. *Bottom*: Cartoon representation of Rpd3L architectural features. See also Supplementary Fig. 7 for primary data relating to positional disorder for Modules I and II. **C** *Left*: Cryo-EM map of the Rpd3L core with individual subunits colored. *Right*: Model of the Rpd3L core with individual subunits colored and labeled.

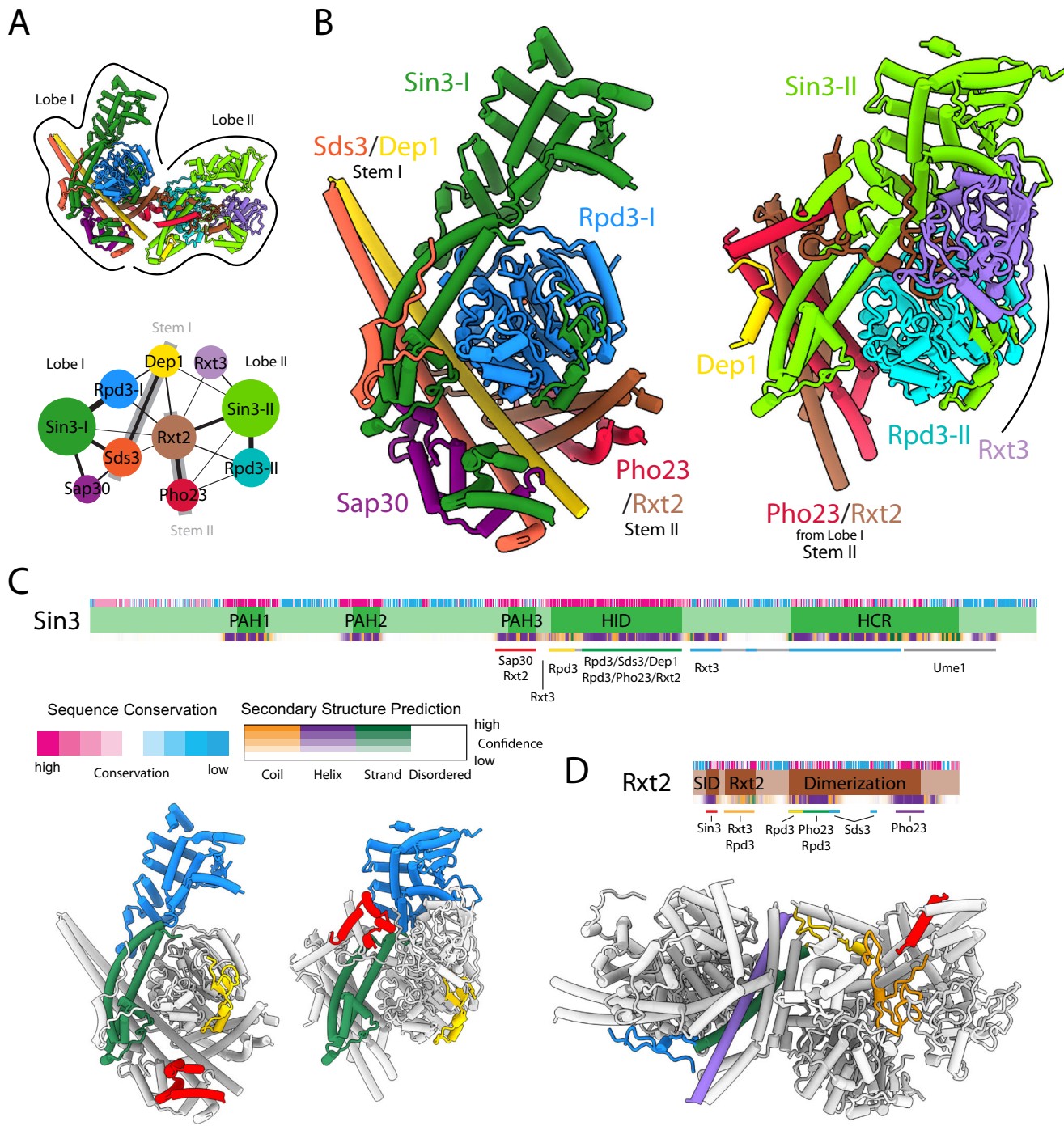

**Fig. 2 | An asymmetric dimeric core in the Rpd3L complex. A** *Top*: Structure of the Rpd3L complex with lobes I and II indicated. *Bottom*: Schematic representation of contacts between various subunits in the complex; the width of each line is proportional to the contact area between subunits. The thick grey lines represent the two stems that formed the corresponding subunits. **B** Rpd3L lobes I (*left*) and II (*right*) aligned based on the Rpd3 subunit. The central four-helix bundle of Rxt2/Pho23 in lobe II plays an analogous scaffolding role to the coiled-coil of Sds3/Dep1 in lobe I. **C** *Top*: Domain map of Sin3 as described in detail in Supplementary Fig. 4. Bars underneath the map represent site(s) of protein–protein interactions with other subunits. *Bottom*: The corresponding regions of Sin3 in the two lobes are shown with the same coloring scheme used for the bars in the top panel. **D** *Top*: Domain map of Rxt2. Bars underneath denote the site(s) of interactions with other subunits. *Bottom*: The corresponding regions of Rxt2 spanning the two lobes are shown with the same coloring scheme used for the bars in the top panel.

(Fig. 2B). The Sin3-Rpd3 interactions are buttressed by extensive interactions with both stems in Lobe I and with Stem II, Rxt3, and other segments of Rxt2 in Lobe II. Sap30 serves to further staple the interaction between Sds3 and Sin3 in Lobe I (Fig. 2B).

Sin3 engages with almost every subunit of the Rpd3L complex, consistent with its expected role as a molecular scaffold for the assembly of the complex (Fig. 2C)[19,20]. Unexpectedly, the Rxt2

polypeptide chain snakes through many nooks and crannies in both lobes, engaging as many as six subunits and serving as a molecular glue while also stitching the two halves of the dimer together (Fig. 2D). Dep1 also spans both lobes, albeit less dramatically, reaching out and engaging Sin3 and to a lesser extent, Pho23, in Lobe II (Fig. 2B).

The differing subunit composition of the stems and the lobes leads to the same regions of Sin3 and Rpd3 contacted by different

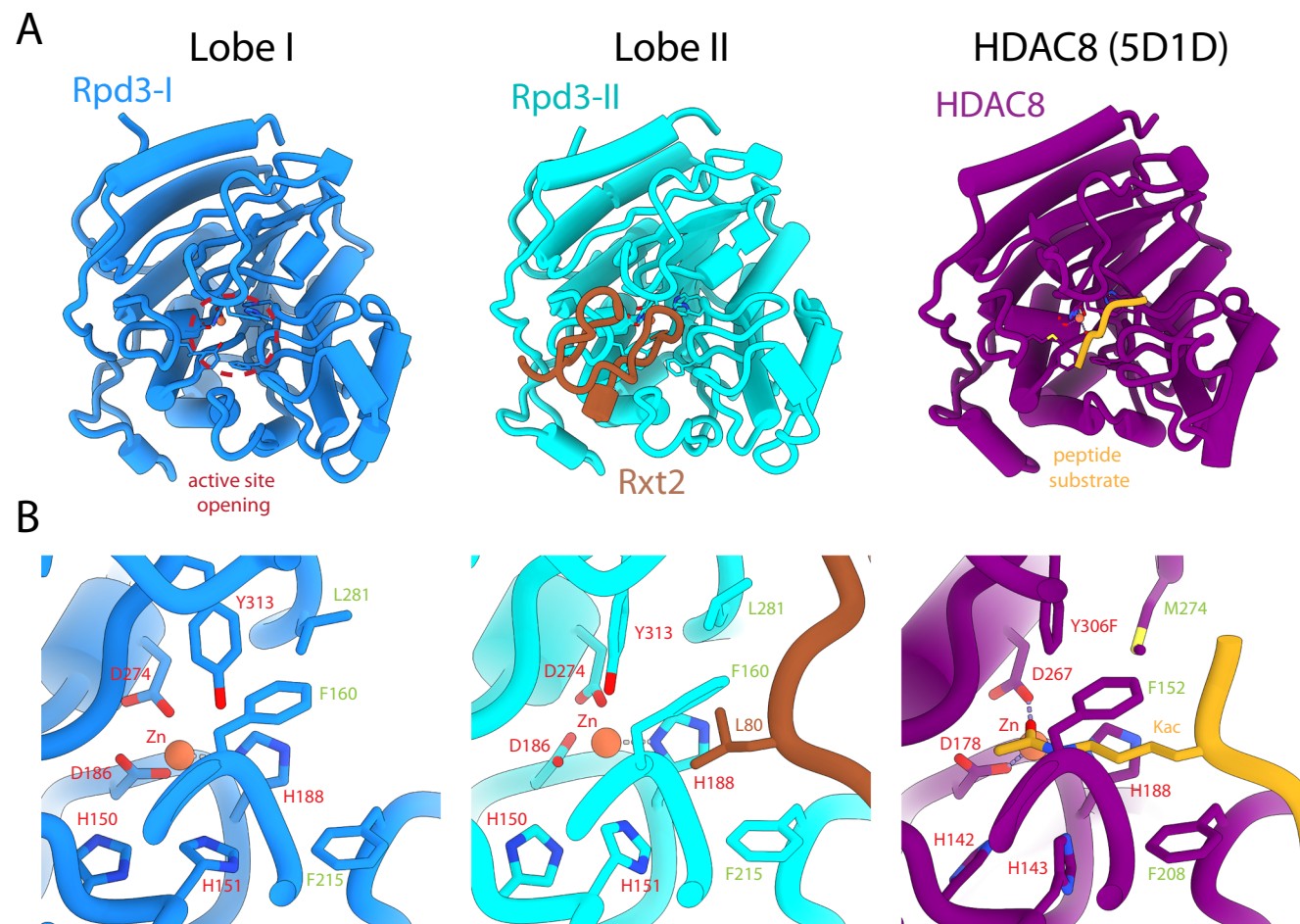

**Fig. 3 | Open and closed active sites of Rpd3 in the two lobes of the Rpd3L complex. A** *Left*: Rpd3 with an open active site (red dashed lines) in lobe I. *Middle*: Rpd3 active site occluded by Rxt2 (brown). *Right*: An HDAC8 mutant (Y306F) bound to an *N*-epsilon-acetyllysine-bearing peptide substrate (gold; PDB ID: 5D1D)[70]. **B** Close-up views of the active sites shown in panel (**A**). Active site residues and the catalytic $Zn^{2+}$ ions are labeled in red, while hydrophobic residues forming the mouth of the active site tunnel are labeled in green for all three proteins. The side chain of Leu80 Rxt2 plugs the opening to the active site tunnel (*middle*), occluding the catalytic site, mimicking the conformation of the *N*-epsilon-acetyllysine at the backbone and side chain levels (*right*).

subunits. For example, whereas the HDAC interaction domain (HID) of Sin3-I is engaged by Sds3, the analogous surface of Sin3-II is involved in interactions with Dep1, with the shared feature being the involvement of a short helix in these interactions (Supplementary Fig. 6A). Interestingly, the PAH3 domain of Sin3-I is engaged by at least three helices of Sap30, but the same domain of Sin3-II is contacted by only a single helix of Rxt2 (Supplementary Fig. 6B). Whereas the so-called highly conserved region (HCR) of Sin3-I is largely devoid of protein-protein contacts, the equivalent region in Sin3-II is engaged to a limited extent by Rxt3 (Supplementary Fig. 6C). Equivalent surfaces of Rpd3 are contacted by the Dep1 and Pho23 subunits forming the two stems, but only one of the helices in these subunits is involved in engaging Rpd3 in both cases (Supplementary Fig. 6D).

Rpd3 is the sole catalytic subunit in the Rpd3L complex. It adopts a strikingly similar conformation in both units of the dimer, like those observed for the mammalian homologues (Fig. 3A). Intriguingly, whereas the active site is readily accessible in Lobe I, the site in Lobe II is completely occluded by a segment of Rxt2. The active site tunnel is blocked near the entrance by Leu80 that mimics the conformation at both the main chain and side chain levels adopted by an *N*$^{\varepsilon}$-acetyllysine substrate bound to an Rpd3 homologue (Fig. 3B). The significance of harboring an inhibited copy of the enzyme is presently unclear, but the pattern and extent of contacts between Rxt2 and Rpd3 suggest that the interface could be disrupted via competitive binding by cognate substrates or conformational changes induced by environmental cues.

## Flexibility and positional disorder in the Rpd3L complex and implications for chromatin engagement

Except for Rpd3, all the subunits in the Rpd3L complex exhibit varying levels of flexibility and positional disorder (Fig. 1A, 1B, & Supplementary Fig. 7). Although as many as eight subunits of the complex contribute towards the ordered core, four of the remaining subunits are either completely invisible or could not be reliably modeled in the experimental cryo-EM maps. Two polypeptides, including the DNA-binding transcription factors Ume6 and Ash1, are invisible because they likely engage with segments of the other subunits (e.g., Sin3 PAH1/2)[21] that are themselves invisible (Fig. 4A, 4B); besides, both Ume6 and Ash1 are present only in sub-stoichiometric amounts relative to the rest of the complex[22]. The solvent-exposed end of Stem I, as well as the tips of the two lobes, exhibit significant flexibility and positional disorder, although these regions also show significant residual density in cryo-EM maps (Fig. 1B & Supplementary Fig. 7). In silico modeling using AlphaFold2-multimer suggests that whereas the former segment comprising Dep1 and Sds3 likely serves as the binding interface for a helical segment at the C-terminus of Cti6, the latter most likely corresponds to a helical domain at the C-terminus of Sin3 that engages with the WD40 domain of Ume1 (Fig. 4A & Supplementary Fig. 7). The significance of the intrinsic flexibility in these regions is unknown but could potentially serve as a mechanism for engagement with diverse chromatin targets since Ume1 is a histone chaperone[23] whereas Cti6 harbors a PHD domain at the *N*-terminus (Fig. 4A). Like Cti6, the Pho23

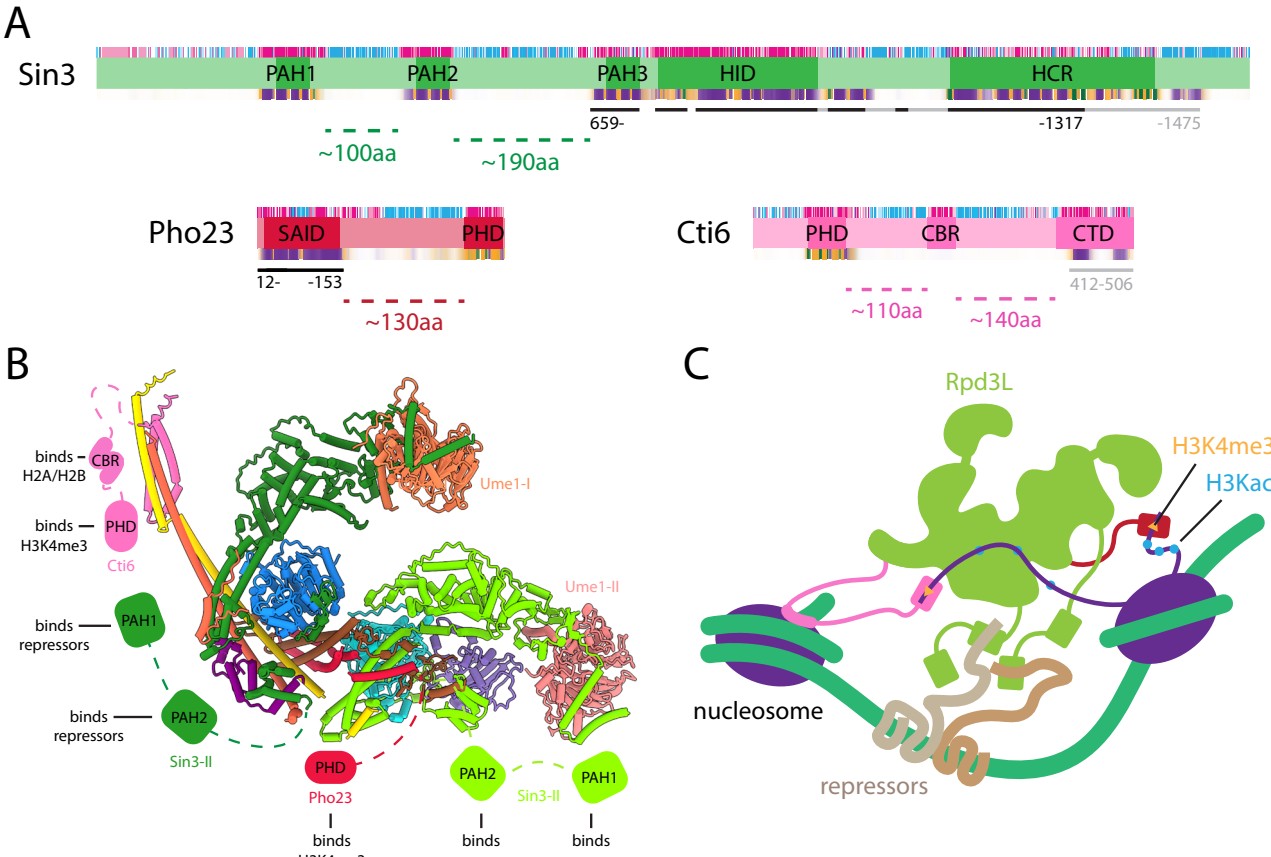

**Fig. 4 | Model of promoter engagement and localized deacetylation by the Rpd3L complex. A** Domain maps of Rpd3L subunits (as described in detail in Supplementary Fig. 4) harboring PAH or PHD domains that are known to engage with DNA-bound repressors or post-translationally modified histones. The bars underneath correspond to polypeptide segments contributing to the structured core of the complex or found within flexible segments (e.g., Cti6). **B** An experimental/AlphaFold2 hybrid model of the Rpd3L complex for the structured core as well as flexibly associated subunits. The PAH and PHD domains of various subunits are rendered as cartoons since they are completely invisible in cryo-EM maps due to flexible tethers connecting these domains with the more structured regions. **C** A cartoon model of the Rpd3L complex showing how this complex could engage with promoter-bound repressors and H3K4me3 containing nucleosomes in the vicinity to perform localized histone deacetylation.

PHD domain is known to bind H3K4me3[24,25] but is also not visible in the structure, again due to the presence of a long, flexible, and poorly conserved tether to the Pho23 segment in the structured core. We surmise that the widespread flexibility allows the complex to be recruited by diverse repressors and function in a variety of promoter and chromatin contexts (Fig. 4C).

Since Cti6 harbors a highly conserved, basic region spanning ~40 residues (designated CBR) within the linker between the PHD domain and the C-terminal helical segment (Fig. 4A), we asked whether a construct containing this region could interact with H2A-H2B heterodimer that harbors the so-called acidic patch in nucleosomes. Since these interactions could be of moderate affinity, we used solution NMR spectroscopy for these studies, as the approach is uniquely suited for these types of interactions. A fusion protein of a yeast H2A-H2B dimer was used as a surrogate for the nucleosome in these studies. As expected, titrations with a 13-residue peptide derived from Kaposi's sarcoma-associated herpesvirus latency-associated nuclear antigen (LANA), an established acidic patch interactor[26], produced large-scale perturbations in the NMR spectrum (Supplementary Fig. 8A). The perturbations produced by the Cti6 construct were also significant but modest compared to the LANA peptide (Supplementary Fig. 8B); however, the perturbations were substantial compared to another peptide derived from a previously described highly conserved, basic region in mouse SDS3[27] that produced little or no changes to the spectrum of the H2A-H2B dimer (Supplementary Fig. 8C). Collectively,

our results suggest that the Rpd3L complex engages chromatin through multivalent interactions mediated by flexible modules with sequence-specific DNA-binding repressors (via Sin3 PAH1/2), H3K4me3 signals (via Pho23/Cti6 PHD), and potentially through the acidic patch (via Cti6; Fig. 4C).

## Comparison with previous genetic, biochemical, and structural studies

A previous mass spectrometric study suggested a two-fold higher abundance in complex stoichiometry for the Rpd3, Sin3, and Ume1 subunits compared to all the other subunits, consistent with our aforementioned results[22]. The same study also assessed the impact of the loss of various subunits on Rpd3L complex integrity and showed that whereas deletion of Rpd3 and Sin3 led to the complete disruption of the complex, loss of the other subunits either produced a graded response or had no effect. For example, the loss of Dep1, Sap30, or Sds3 was also extremely disruptive but did not preclude Rpd3 from assembling with Sin3 and Ume1, whereas the loss of Rxt2, Rxt3, or Pho23 was disruptive to the sub-assembly of these three subunits but did not preclude Rpd3 from assembling with Sin3, Ume1, Dep1, Sds3, and Sap30, consistent with our structure. This suggests that the scaffolding roles of Pho23 and Rxt2 are critical for proper Lobe II assembly, whereas the architectural roles played by Sds3 and Dep1 are vital for preserving the overall integrity of the complex (Fig. 2). In other words, Lobe II cannot exist without Lobe I whereas the Lobe I can assemble without

Lobe II. On the other hand, the loss of Cti6, Ash1, and Ume6 produced the least impact on complex integrity, again in line with their peripheral association with the complex, and consequently, their non-involvement in complex assembly. Similar results as above were also obtained by other genetic and biochemical studies[6,28–31]. Significantly, in one study, the loss of Pho23 or Rxt2 had no effect on the overall deacetylase activity of the complex compared to wild-type, in contrast to the significant reductions in activity caused by the loss of either Dep1 or Sds3[6]. This suggests that Rpd3-II, which is inhibited in our structure (Fig. 3), might not contribute significantly towards the overall deacetylation activity of the complex, at least under normal conditions.

All previous structural studies relating to Rpd3L complex assembly were described for various subunits of the corresponding mammalian complex. The NMR and crystal structures of various sub-complexes, including those of SIN3A and SDS3[27], SIN3A and SAP30[32], as well as the SDS3 homodimer[27] all show backbone root-mean-square deviations in the 0.73–0.90 Å range with our structure. Previous structure-function studies of the interactions between SIN3A and HDAC1[27,33], SIN3A and SDS3[27,34], SIN3A and SAP30[32,35], as well as SDS3 and BRMS1[27] are also entirely consistent with our structure. However, these structural and biochemical studies focused only on the major determinants of these interactions; our cryo-EM structure reveals additional interactions governing the higher-order folding of the complex, thereby providing unprecedented insights into how the complex is assembled and organized. The overall architecture of the yeast complex is also broadly consistent with previous crosslinking mass spectrometric analyses of the mammalian complex, thereby providing additional independent support for our structure[36]. Previous studies anticipated a symmetric complex based on the presence of multiple dimerization motifs[27]; however, as vividly illustrated in our structure, these motifs mediate heterodimerization and perform important scaffolding functions for assembly and higher-order folding of the complex (Fig. 2B).

## Implications for the mammalian Sin3L/Rpd3L complex and the yeast Rpd3S complex

The high level of sequence and structural conservation shared by key subunits of the yeast Rpd3L complex with the mammalian counterpart suggests that the two complexes likely share grossly similar features. Whereas orthologs for Sin3 (SIN3A/B), Rpd3 (HDAC1/2), Ume1 (RBBP4/7), Sds3 (SDS3), SAP30 (SAP30), Pho23 (ING1b/2), Dep1 (BRMS1) are either well-established or strongly suspected, knowledge of the Rpd3L complex structure combined with AlphaFold2-multimer predictions allowed us to hypothesize that SAP130a/b and ARID4A/B in the corresponding mammalian complex play structurally equivalent roles as Rxt2 and Cti6, but as analogs (Supplementary Fig. 9A–C). These predictions were then tested by protein–protein interaction assays. The orthologous/analogous domains of SDS3, BRMS1, and ARID4A, as well as SAP130 and ING2, were expressed or co-expressed in E.coli purified and tested for the formation of stable complexes in solution by size-exclusion chromatography. The corresponding recombinant proteins co-eluted as predicted by the AlphaFold models (Supplementary Fig. 9D), establishing homology/analogy for all nine subunits shared by the yeast and mammalian complexes.

The five-subunit yeast Rpd3S complex shares three subunits, including Rpd3, Sin3, and Ume1, with the Rpd3L complex. Since these subunits assemble to form a sub-complex in the Rpd3L complex, contributing substantially to the total mass of each lobe, we anticipate the structure of this part of the complex to be identical to the Rpd3S complex. Indeed, the presence of two copies of these subunits in the Rpd3L complex provides two independent views of the sub-complex (Fig. 2B). The structured core within the sub-complex comprising the Rpd3 and Sin3 subunits adopts almost identical conformations in the two lobes, notwithstanding the disparate contacts made by these subunits with the other subunits in each lobe.

In conclusion, we have described the cryo-EM structure of the yeast Rpd3L complex, an evolutionarily conserved HDAC complex that is widely distributed in eukaryotes. The complex assembles as an asymmetric dimer with two copies of the sole catalytic subunit that is tightly integrated into the complex through the actions of an overwhelming majority of subunits that perform critical scaffolding functions. Our studies suggest that the corresponding mammalian Sin3L/Rpd3L complex most likely shares broadly similar overall features, although some innovations are expected due to molecular evolution. Our structure of the yeast Rpd3L complex provides a foundation for deeper mechanistic studies in model organisms as well as humans and also for the design and development of HDAC complex-specific inhibitors as novel therapeutics.

## Methods

### Protein purification

The Rpd3L complex was purified from Saccharomyces cerevisiae using a tandem affinity purification (TAP) method as described[37,38]. A strain modified with a TAP tag at the C-terminus of Rxt2 was obtained from GE Dharmacon and grown at 30 °C in YPD. A six-liter culture of this strain was grown to stationary phase and harvested at $OD_{600}$ of 6, resuspended in 180 ml of lysis buffer (100 mM HEPES pH 7.9, 500 mM KCl, 2 mM $MgCl_2$, 10% glycerol, 500 mM PMSF, 500 mM 1,10-phenanthroline, 2.2 mM pepstatin A, 5 mM E-64, 5 mM leupeptin). Cells were lysed using a beat beater with a total process time of 15 min alternating between 30 s ON and 60 s OFF. The lysate was filtered through a Miracloth (Millipore) and transferred to a beaker to which 15 µl of DNase and Rnase (both 5 mg/ml; Thermo Fisher Scientific) along with NP-40 to a final concentration of 0.02% were added and stirred at 4 °C for 10 min. Another 15 µl of Dnase and Rnase were added, and the contents were stirred for an additional 10 min. 75 mg of heparin powder (Sigma) dissolved in water was added to the lysate and stirred for 10 min. The lysate was then clarified by centrifugation at 15,000 g for 60 min. The supernatant was collected and passed through a filter before incubation with gentle rocking with 0.8 ml packed IgG resin (Cytiva) for 4 h at 4 °C. After incubation, the resin was collected by centrifugation at 2000 g for 5 min and the supernatant was discarded. The resin was transferred to a 1.5 ml microcentrifuge tube and washed 5 times with lysis buffer and 10 times with TAP buffer (20 mM HEPES pH 7.9, 2 mM $MgCl_2$, 10% glycerol, 500 mM KCl, 1 mM TCEP, 0.02% NP-40). The resin was then resuspended in an equal volume of TAP buffer with 25 µg of TEV protease and incubated with end-over-end rotation in the cold room overnight. Following cleavage, samples were spun to collect the supernatant. Calcium chloride was added to the supernatant to a final concentration 2 mM before incubation for 4 h with 100 µl of calmodulin resin (Cytiva) that had been pre-equilibrated with TAP buffer containing 2 mM $CaCl_2$. The supernatant was removed and the resin was washed with two column volumes of TAP buffer containing 750 mM KCl and 2 mM $CaCl_2$, followed by three column volumes with TAP buffer containing 250 mM KCl and 2 mM $CaCl_2$, followed by two column volumes with TAP buffer containing 250 mM KCl. Protein was eluted from the resin with the addition of 200 µl of elution buffer (20 mM HEPES pH 7.9, 2 mM $MgCl_2$, 10% glycerol, 250 mM KCl, 5 mM EGTA, 1 mM TCEP, 0.01% NP-40) which was incubated at 4 °C for 30 min with end-over-end rotation. A total of six elutions were collected and analyzed by SDS-PAGE stained with Flamingo staining solution (Bio-Rad). Samples were concentrated 10-fold, aliquoted, snap-frozen in liquid nitrogen, and stored at −80 °C.

A maltose-binding protein (MBP)-tagged human BRMS1 (residues 51–179) construct in the pMCSG23 vector was co-expressed with a His6-tagged mouse SDS3 (residues 43–234) construct in the pMCSG7 vector in E. coli at 16 °C and was combined with His6-tagged human ARID4A R2 domain (residues 1164–1230) construct in the pMCSG7 vector that was separately expressed in E. coli at the same temperature[39]. The proteins were purified using $Ni^{2+}$-NTA affinity chromatography as

described previously[40]. The purified proteins were then concentrated and subjected to size-exclusion chromatography (SEC) using a Superdex 75 GL column (GE Healthcare) and a running buffer comprising 25 mM MOPS (pH 7.5), 150 mM NaCl, and 1 mM TCEP. An MBP-tagged human SAP130 (residues 924–1048) construct in the pMCSG23 vector was co-expressed with a $His_6$-tagged human ING2 SAID domain (residues 18–131) construct in the pMCSG7 vector *E. coli* at 16 °C and purified using $Ni^{2+}$-NTA affinity chromatography followed by SEC as described above.

A $His_6$-tagged construct in pMCSG7 vector of a yeast H2B-H2A.Z fusion protein was generated using the pET-TBZ construct as the PCR template (plasmid #28239 from Addgene)[41]. The protein was expressed in minimal media supplemented with $^{15}$N-ammonium sulfate and glucose in *E. coli* BL21(DE3) cells harboring the pRARE2 plasmid at 16 °C. The protein was purified using $Ni^{2+}$-NTA affinity chromatography as described above except all buffers contained 6 M guanidine hydrochloride (Gdn.HCl). The protein was refolded in buffer containing 0.5 M NaCl, followed by TEV protease cleavage overnight in buffer containing 0.2 M NaCl, followed by purification of the cleaved protein using reversed phase HPLC. The protein was then lyophilized, resuspended in 6 M Gdn.HCl and refolded using the two-step protocol above for NMR studies. A $His_6$-tagged construct in pMCSG7 vector of yeast Cti6 (residues 221-290) containing the CBR and a GST-tagged construct in pMCSG10 vector of mouse SDS3 (residues 172–209) were expressed in pRARE2 containing *E. coli* at 16 °C. The proteins were purified under native conditions using $Ni^{2+}$-affinity chromatography as described above, cleaved with TEV protease overnight to remove the tags, and the cleaved proteins were purified using reversed phase HPLC. The 13-residue LANA peptide was synthesized using automated methods in the Simpson Querry Institute Peptide Core facility at Northwestern and the crude was purified by reversed phase HPLC. The identity of all proteins and peptides were verified by mass spectrometry prior to use.

### Negative stain EM sample preparation, data collection and data processing

Purified Rpd3L complex was diluted 1:3 with crosslinking buffer (20 mM HEPES pH 7.9, 0.1 mM EDTA, 2 mM $MgCl_2$, 1% glycerol, 75 mM KCl, 1 mM BS3 (Thermo Fisher Scientific)) and allowed to crosslink on ice for 15 min. A total of 4 µl was then applied to a glow-discharged continuous carbon grid for 5 min and stained using uranyl formate. The negative stain dataset was collected on a JOEL 1400 microscope (JOEL) operating at 120 keV and equipped with an Ultrascan 4000 camera (Gatan) at a pixel size of 1.855 Å/pixel. Data were collected using Leginon data acquisition software[42]. Initial 2D classification was performed in Appion[43–45]. A 3D ab initio model was generated using Relion (version 3.1)[46]. The contrast transfer function (CTF) parameters were estimated using Gctf (version 0.5) and particles were picked using Relion[46,47]. Extracted particles were subjected to 2D classification, ab initio model generation, and 3D classification.

### Cryo-EM sample preparation, data collection and data processing

Cryo-EM samples were prepared using Vitrobot Mark IV (FEI). The Rpd3L complex was crosslinked on ice using 1 mM BS3 (Thermo Fisher Scientific) for 15 min before 4 µl of the sample was applied to a graphene oxide coated 2/1 Quantifoil grid (Quantifoil) at 4 °C under 100% humidity[48,49]. The sample was immediately blotted using Whatman #1 for 1.5 s at 0 N force and then immediately plunge frozen in liquid ethane cooled by liquid nitrogen.

Grids were clipped and transferred to the autoloader of a Titan Krios electron microscope (Thermo Fisher Scientific) operating at 300 keV acceleration voltage. Images were recorded with a post energy filter K3 direct electron detector (Gatan) operating in super-resolution mode at a calibrated magnification of 60,532 (0.413 Å/pixel), using the SerialEM data collection software[50]. 45-frame exposures were taken at 0.093 s/frame, using a dose rate of 9.75 e⁻/pixel/s (1.33 e⁻/Å²/frame), corresponding to a total dose of 60 e⁻/Å² per micrograph. A total of 4486 movies were collected at 0° tilt and 9615 movies were collected at 40° tilt from 1 grid.

Cryo-EM data processing was performed using Relion 4.0[51]. Whole movie frames were aligned to correct for specimen motion. The CTF parameters were estimated using Gctf[47]. 2,220,934 particles were picked using Relion LoG picker, extracted and subjected to three rounds of 2D classification (EM). Seven 2D class averages were low pass filtered to 20 Å and used for template picking, resulting in 2,547,251 particle picks. Particles were extracted binned by 12 and subjected to three rounds of VDAM two-dimensional classification and one round of EM two-dimensional classification to remove ice, empty picks, and graphene oxide creases, which resulted in 210,993 particles[51]. Selected particles were re-extracted binned by 4 subjected to three-dimensional refinement. The refined particles were subjected to three rounds of CTF refinement, particle polishing and three-dimensional refinement[46,52]. The resulting structure attained a gold standard resolution of 3.5 Å, although 3D Fourier Shell Correlation (FSC) analysis reveals a high amount of anisotropy (Supplementary Fig. 2)[53]. To further improve the map quality, Lobes I and II were separately focus refined and both resulted in a resolution of 3.4 Å. All resolution calculations were determined from gold-standard refinements at an FSC of 0.143[54,55].

### Model building and refinement

AlphaFold2-multimer predictions were computed via the ColabFold interface for each pair of Rpd3L complex components[56,57]. From these predictions select 3- and 4-component predictions were computed. The resulting models were used to dock into focus refined maps of Lobes I and II of Rpd3L (Fig. 1C). Docked models were manually trimmed, extended, and modified to fit the maps using COOT[58]. In addition, ModelAngelo was performed for both maps, resulting in the identification of a section of difficult-to-identify Rxt2 density[59]. This region was also manually trimmed, extended, and modified to fit the maps using COOT. The resulting protein model was iteratively refined using PHENIX and manually adjusted in COOT[60,61]. The model was validated using MTRIAGE and MOLPROBITY within PHENIX[62,63]. The refinement statistics show values typical for structures in this resolution range (Table 1). The FSC curve between the model and the map shows good correlation up to 3.9 Å resolution according to the FSC = 0.5 criterion[55].

An extended Rpd3L model, as seen in (Fig. 4B), has been provided as Supplementary Data 1. (1) The extended model includes the refined Rpd3L core along with the AlphaFold2 extensions for Dep1, Sds3 and Sin3 and models for Cti6 and two copies of Ume1.

### HDAC assays

Deacetylase assays were performed using a model acetylated peptide substrate as previously described[64]. The concentration of the Rpd3L complex used in these assays were estimated based on SDS-PAGE band intensities of various subunits following Coomassie Brilliant Blue staining compared to known amounts of bovine serum albumin; band intensities were quantified using AzureSpot software (Azure Biosystems). Assay data were fitted using GraphPad Prism ver. 9.5.0 (Dotmatics).

### NMR spectroscopy

NMR spectra were recorded on a 600 MHz Bruker Neo spectrometer equipped with a QCI-F cryoprobe at 308 K. NMR samples were prepared in 20 mM MES buffer (pH 6.0) containing 200 mM NaCl. Initial $^{15}$N-H2B-H2A.Z sample concentrations for the titrations ranged between 95 and 170 µM; titrations with each of the three peptides were conducted until four molar equivalents had been added. $^{1}$H-$^{15}$N TROSY spectra were recorded with 16 scans, 128 complex increments in the

**Table. 1 | Data Collection, Map and Model Refinement and Validation Statistics**

| Data Collection | |
|---|---|
| Dataset | Rpd3L Complex |
| Microscope | Titan Krios |
| Stage type | Autoloader |
| Voltage (keV) | 300 |
| Detector (Mode) | Gatan K3 (super-resolution) |
| Pixel size (Å) | 0.826 (0.413) |
| Defocus range[μm (mean/STD)] | 2.22 (0.46) |
| Electron dose (e⁻/Å²) | 60 |
| **Reconstruction** | |
| Software | RELION |
| Particles | 210,993 |
| Box size (pixels) | 272 |
| Accuracy rotations (°) | 1.929 |
| Accuracy translation (pixels) | 0.999 |
| Map resolution (Å) | 3.5 |
| Map sharpening b-factor (Å²) | −57 |
| **Coordinate refinement** | |
| Software | PHENIX |
| Resolution cutoff (Å) | 3.5 |
| **Model** | |
| Residues | |
| Protein | 2675 |
| B-factors | |
| Protein | 67.42 |
| RMS deviations | |
| Bond lengths (Å) | 0.003 |
| Bond angles (°) | 0.56 |
| **Validation** | |
| Molprobity Score | 1.70 (89th) |
| Molprobity clashscore | 11.05 (67th) |
| Rotamer outliers (%) | 0.29 |
| Cβ deviations (%) | 0 |
| Ramachandran plot | |
| Favored (%) | 97.25 |
| Allowed (%) | 2.75 |
| Outliers (%) | 0 |

indirect $^{15}$N dimension and spectral widths of 16 and 30 ppm for the $^{1}$H and $^{15}$N dimensions, respectively. Spectra were processed using Top-Spin ver. 4.2.0 (Bruker) and analyzed using NMRFAM-Sparky ver. 1.470[65].

## Creation of figures
Molecular images were generated using UCSF ChimeraX[66]. Protein domain maps were generated using domainsGraph.py (https://github.com/avibpatel/domainsGraph)[67–69].

## Reporting summary
Further information on research design is available in the Nature Portfolio Reporting Summary linked to this article.

## Data availability
Cryo-EM density maps have been deposited in the Electron Microscopy Data Bank (EMDB) under accession numbers EMD-29892 (overall Rpd3L complex), EMD-29876 (lobe I), and EMD-29875 (lobe II). Model coordinate of the Rpd3L complex has been deposited in the Protein Data Bank (PDB) under accession numbers 8GA8. Source data are provided as a Source Data file. Source data are provided with this paper.

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

## Acknowledgements

We thank Jason Pattie for computer support, Janette Meyers at the PNCC for data collection support and Yongbo Zhang for assistance with NMR experiments. We are grateful to the National Institutes of Health and the Robert H. Lurie Comprehensive Cancer Center for supporting structural biology and proteomics research at Northwestern (P30 CA060553, S10 OD025194, and P41 GM108569). The studies described herein were supported in part by the National Institutes of Health (R01GM135651, R01GM144559, and P01CA092584 to Y.H.), the American Heart Association (17GRNT33680167 to I.R.), the H Foundation, and the Sherman Fairchild Foundation (I.R.) and the Baker Program in Undergraduate Research at Northwestern (K.H.T., S.Z., M.L., and I.R.). A portion of this research was supported by NIH grant U24GM129547 and performed at the PNCC at OHSU and accessed through EMSL (grid.436923.9), a DOE Office of Science User Facility sponsored by the Office of Biological and Environmental Research.

## Author contributions

A.P., I.R. and Y.H. conceived the project. A.P. performed most of the experiments, analyzed cryo-EM data and built the models. J.Q., S.Z., and M.L. contributed to protein purification and biochemical characterizations; K.H.T. performed protein purifications and NMR spectroscopic characterizations. A.P., I.R. and Y.H. wrote the manuscript.

## Competing interests

The authors declare no competing interests.
