## [Peer Review File · Nature Communications]

Cryo-EM Structure of the *Saccharomyces cerevisiae* Rpd3L Histone Deacetylase ComplexREVIEWER COMMENTS

Reviewer #1 (Remarks to the Author):

In this work, the authors analyse the structure and molecular mechanisms of the *S. cerevisiae* Rpd3L complex, one of the two major histone deacetylase-containing transcriptional repression complexes in budding yeast.

Using cryo-electron microscopy with merged titled single particle analysis datasets, the complex was resolved to 3.5 Å and 3.4 Å in a subsequent focused classification approach. Structural elucidation by the molecular modeling of the Rpd3L Coulomb potential maps reveals the function of its various subunits. The catalytic center, Rpd3, exists in both cores of the two asymmetric dimeric lobes of the complex. The other subunits act chiefly as scaffolding and sterically block one Rpd3 center within the isolated complex.

Rpd3L complex had so far not been structurally analysed at molecular resolution. These findings are furthermore significant in that Rpd3L's subunits show a surprising structural resemblance to unrelated subunits found in fungal and mammalian HDAC complexes, underlining their common mechanistic function across the domains of life.

The article is well written, clearly structured and no major issues were found by this reviewer.

Methodologically, the authors were faced with a preferred orientation bias of the 3D-reconstructed map, which they addressed through the recording and merging of a 40° stage-tilted dataset (Supplementary Figure S2). Based on the Euler angle plots, preferred orientation is still significant, though the resolved density seems adequately interpretable.

Some minor corrections / remarks:

Line 59: "2D class averages **of particles picked** from electron micrographs [...]"

Line 275: It would be useful to add magnification factor/pixel size information to the negative stain protocol.

Lines 296/297: "Whole movie frames were aligned to correct for specimen motion [...]": Was no patch-tracking performed for motion correction?

In summary, this work presents a novel, well-executed analysis of an HDAC structure at close to molecular resolution. Recommended for publication.

Reviewer #2 (Remarks to the Author):

This manuscript describes the CryoEM based structure of the yeast RPD3L complex. In this body of work the authors used the Rxt2 subunit, which is part of the RPD3L complex, to affinity purify complexes from yeast. Complexes crosslinked with BS3 were then analyzed using a Titan Krios electron microscope and from this a model for the RPD3L complex was built. The RPD3L complex is an important and conserved chromatin remodeling complex for which little structural information has existed prior to this work. The manuscript is generally well written and provides a detailed analysis of the structure. As such, this manuscript makes an important contribution to the scientific literature and publication is fully warranted in Nature Communications after minor revisions.

To begin, what would greatly strengthen this manuscript is some mutational analysis to prevent complex assembly. The authors cite a manuscript where entire proteins were deleted from the yeast genome and protein complex assembly studied, but here they have finer details regarding the specific sequences and domains from specific proteins that one would hypothesize would disrupt the complex. For example, Figure 1 seems to suggest that Pho23 and Rxt2 are particularly important for holding the complex together. Data showing that deleting specific regions of each protein disrupt the complex would greatly strengthen the manuscript. The other possibility would be to do the same types of experiments for Sds3 and Dep1 to show deletion of specific regions of these proteins affect the complex rather than the prior work cited where whole proteins were deleted.

Next, the authors comparison of this structure to the human complexes would benefit from further discussion. A more detailed analysis comparing and contrasting the proposed structure in reference 35 would be very useful. In the current version of this manuscript the authors simply state that these structures are 'broadly consistent' but do not provide a detailed analysis to support this statement. A more rigorous statistical/mathematical/alignment-based comparison is needed to improve this section of the manuscript.

Another area that the manuscript could be improved is expanding on how the complex might engage with chromatin and potentially providing some experimental evidence to support this. In Figure 6 they begin to cover this, but only a rough model is presented where some of the domains of specific proteins are highlighted that are known to have specific chromatin interactions. Here, the manuscript would greatly benefit from a more detailed discussion of the prior literature for these domains, for example. The authors should then discuss whether or not the specific regions, domains, and key amino acids mediating these interactions are accessible or buried in the structure that they present.

Lastly, the manuscript would also benefit from an analysis of the experimental molecular weight of the complex they built the model on and compare it with the density map they obtained. This could be done in the supplement but Figure 1B looks to have a sizable amount of unassigned density and speculating

on what might be in these areas is warranted and would also improve the manuscript. In figure 4 they appear to be fitting Ume1 into these areas, but a more detailed analysis and explanation would be helpful.

Reviewer #3 (Remarks to the Author):

In this study, Patel et al. report the cryo-EM structure of the chromatin-associated yeast Rpd3L complex. Using TAP tag-mediated purification of the native yeast complex through the Rxt2 subunit, they isolate all canonical subunits of the complex and acquire and process cryo-EM data to a 3.5 Å resolution. Their analysis confirms previous studies reporting a two-lobe confirmation of the complex, each lobe containing the catalytic subunit Rpd3, as well as the Sin3 and Ume1 proteins. These results are also consistent with the previously described stoichiometry of the complex. Interestingly, they demonstrate that most components of the complex serve as scaffolding subunits, several of them exhibiting a flexible conformation, and extensive intra-complex surface contacts. Such conformation would make the stable complex amenable to recruitment by transcription factors and tethering to chromatin. The most important finding may be the discovery that one of the two Rpd3 subunits of the complex, namely the one associated with the Rxt3 lobe, is likely catalytically inactive as a result of steric hindrance from the Rxt2 subunit. Altogether, the results presented are clear and convincing. While the study mostly confirms previous models of the Rpd3L complex, it also provides novel insight on the respective contribution of each subunit of the complex, and by extension, of the mammalian homologs of these components.

Some concerns mitigate the enthusiasm for the study.

First, there is little attempt to explain known activities of specific subunits of the complex based on the structure elucidated here. For example, the authors do not speculate in depth on the molecular bases underlying the requirement for Sds3 in Rpd3L deacetylase activity. A more throughout discussion of the gain in knowledge obtained from the structure as it relates to the previously established properties of some of the subunits would be warranted.

Second, a major part of the results obtained here is largely confirmatory of previous models of the Rpd3L complex structure. In particular, the study by Sardiù et al, PLoS ONE has suggested that the Rpd3L complex structure consisted of two asymmetric modules, each containing Rpd3 and Sin3 components. Given the striking discovery that one of the two Rpd3 subunits may be inactive, a deeper understanding of this feature would greatly increase the impact of this study. It would thus be helpful for the authors to put forward and experimentally test a model that would justify the need for this inactive lobe in the Rpd3L canonical complex, or at least experimentally confirm that the presence of Rxt2 inactivates lobe2-Rpd3.

Response to Reviewers' Comments

We are pleased that all the reviewers thought our work was important and impactful to warrant publication in *Nature Communications*. We thank them for their thoughtful and constructive critiques. Below, we reproduce their comments in full and respond to their concerns to the best of our abilities (the reviewers' concerns are highlighted in green and our responses are in blue).

Reviewer #1 (Remarks to the Author):

In this work, the authors analyse the structure and molecular mechanisms of the *S. cerevisiae* Rpd3L complex, one of the two major histone deacetylase-containing transcriptional repression complexes in budding yeast. Using cryo-electron microscopy with merged titled single particle analysis datasets, the complex was resolved to 3.5 Å and 3.4 Å in a subsequent focused classification approach. Structural elucidation by the molecular modeling of the Rpd3L Coulomb potential maps reveals the function of its various subunits. The catalytic center, Rpd3, exists in both cores of the two asymmetric dimeric lobes of the complex. The other subunits act chiefly as scaffolding and sterically block one Rpd3 center within the isolated complex.

Rpd3L complex had so far not been structurally analysed at molecular resolution. These findings are furthermore significant in that Rpd3L's subunits show a surprising structural resemblance to unrelated subunits found in fungal and mammalian HDAC complexes, underlining their common mechanistic function across the domains of life.

The article is well written, clearly structured and no major issues were found by this reviewer.

Methodologically, the authors were faced with a preferred orientation bias of the 3D-reconstructed map, which they addressed through the recording and merging of a 40° stage-tilted dataset (Supplementary Figure S2).

Based on the Euler angle plots, preferred orientation is still significant, though the resolved density seems adequately interpretable.

Some minor corrections / remarks:

Line 59: "2D class averages **of particles picked** from electron micrographs [...]"

We thank the reviewer for pointing this out; we have made correction to the text.

Line 275: It would be useful to add magnification factor/pixel size information to the negative stain protocol.

We thank the reviewer again for bringing this inadvertent omission to our attention; we have now added this information to the text.

Lines 296/297: "Whole movie frames were aligned to correct for specimen motion [...]": Was no patch-tracking performed for motion correction?

Yes, this is correct. Patch tracking was not performed to enable faster processing. We performed RELION particle polishing instead, so local motion was corrected, albeit *after* particle sorting.

In summary, this work presents a novel, well-executed analysis of an HDAC structure at close to molecular resolution. Recommended for publication.

Reviewer #2 (Remarks to the Author):

This manuscript describes the CryoEM based structure of the yeast RPD3L complex. In this body of work the authors used the Rxt2 subunit, which is part of the RPD3L complex, to affinity purify complexes from yeast. Complexes crosslinked with BS3 were then analyzed using a Titan Krios electron microscope and from this a model for the RPD3L complex was built. The RPD3L complex is an important and conserved chromatin remodeling complex for which little structural information has existed prior to this work. The manuscript is generally well written and provides a detailed analysis of the structure. As such, this manuscript makes an important contribution to the scientific literature and publication is fully warranted in Nature Communications after minor revisions.

To begin, what would greatly strengthen this manuscript is some mutational analysis to prevent complex assembly. The authors cite a manuscript where entire proteins were deleted from the yeast genome and protein complex assembly studied, but here they have finer details regarding the specific sequences and domains from specific proteins that one would hypothesize would disrupt the complex. For example, Figure 1 seems to suggest that Pho23 and Rxt2 are particularly important for holding the complex together. Data showing that deleting specific regions of each protein disrupt the complex would greatly strengthen the manuscript. The other possibility would be to do the same types of experiments for Sds3 and Dep1 to show deletion of specific regions of these proteins affect the complex rather than the prior work cited where whole proteins were deleted.

We thank the reviewer for these suggestions. We would like to note that our results are also consistent with the results from numerous site-specific mutations in the SDS3 (Sds3 ortholog), BRMS1 (Dep1 ortholog), SAP30, Sin3A, and HDAC1 subunits described in our previous structure-function studies of the orthologous mammalian complex (we cite these studies in the section “Comparison with Previous Genetic, Biochemical, and Structural Studies”. We also note that in the original manuscript we tested via size exclusion chromatography which specific domains of SDS3, BRMS1, and ARID4A (Cti6 analog), and similarly, ING2 (Pho23 ortholog) and SAP130 (Rxt2 analog) were involved in interactions analogous to those in the yeast complex (**Supp. Fig. 9**). We believe that these are better tests of the hypotheses generated by our structural model, as some of these interactions were not anticipated and involve mammalian analogs that obviously share little or no sequence similarity with the corresponding yeast proteins.

We note that our structural characterizations were performed following biochemical isolation of the endogenous Rpd3L complex, which is present in low abundance in yeast. Since the purified complex represents the native complex, that we showed is active and responsive to inhibitors in new data (**Supp. Fig. 1C & 1D**), and since the cryo-EM map is at 3.4 Å resolution, there was little ambiguity in assigning various regions of the map to the constituent subunits. We exercised considerable care in not overinterpreting the cryo-EM maps.

Next, the authors comparison of this structure to the human complexes would benefit from further discussion. A more detailed analysis comparing and contrasting the proposed structure in reference 35 would be very useful. In the current version of this manuscript the authors simply state that these structures are ‘broadly consistent’ but do not provide a detailed analysis to support this statement. A more rigorous statistical/mathematical/alignment-based comparison is needed to improve this section of the manuscript.

We would like to first clarify some misconceptions about previous structural models proposed for this complex. As far as we are aware, there have been only two such models proposed based on experimental data – one, our own previous work that suggested a symmetric dimer (ref. 30) and another that makes no explicit references to the dimerization state of the complex (ref. 35). The former model was derived following reconstitution and

structural analysis of much smaller sub-complexes of the mammalian/human Sin3L complex, whereas the latter model was derived using sparse inter-subunit cross-linking mass spectrometry data following affinity purification of the complex. Inter-subunit cross-links were cataloged for 5 subunits and the model was built using 9 restraints for 3 subunits. The figure below shows our structure on the left and the model proposed in ref. 35 on the right. Homologous subunits are colored using the same scheme (HDAC1/Rpd3: blue, SIN3A/Sin3: green, SAP30/Sap30: purple). As is readily evident, the two models are substantially different and we did not think that including this comparison would be meaningful, especially given how the two models were generated.

The differences between the models notwithstanding, we did find that the cross-linking data was “broadly consistent” with our structure. To perform this comparison, we generated an AlphaFold2-guided model for the human complex by matching homologous segments of the yeast and human complexes, as shown in Supplementary Figure S8. This model was then used to map the cross-linking data from ref. 35. As shown in the figure below, 71% of the cross-links mapped for the holo complex (*left*) and 89% of the cross-links for the core complex (*right*) were within 30 Å (connectivities that are <30 Å are colored blue while those that are >30 Å are colored red). Most of the cross-links that were mapped to be >30 Å were between the SIN3A HCR domain and other parts of the complex. This we suspect is due to this domain adopting either a different orientation in the human complex (than in the yeast complex) or that this domain is flexible and can generate cross-links with many different parts of the complex that are (>30 Å away). An example of a protein complex that is homologous at the sequence level (between yeast and human) but results in different overall shapes for the corresponding complexes would be the SAGA complex (compare: <https://www.nature.com/articles/s41586-020-1933-5> with <https://www.nature.com/articles/s41594-021-00682-7>). In the SAGA complex, the major modules/lobes of the complex adopt similar structures between yeast and human, but the arrangements of these modules/lobes are significantly different. We surmise that something similar may be happening for the yeast Rpd3L and human SIN3A complexes. The other plausible explanation for the large number of crosslinks with >30 Å mapped distance could be that the SIN3A HCR domain adopts alternative conformations. This is exemplified in the TFIID (<https://www.science.org/doi/full/10.1126/science.aau8872>) and PI3-Kinase complexes (<https://doi.org/10.1016/j.molcel.2017.07.003>), where crosslinks >30 Å were readily explained by the alternative conformations adopted by the complexes.

Finally, we note that the human complex features new subunits and other innovations including new domains attributable to molecular evolution that are absent in the yeast complex, suggesting that these differences could also contribute to the overall shape and organization of the human complex.

We did not elaborate on these points in the original manuscript given the multi-faceted nature of this comparison that we thought would detract from the key findings described in the manuscript. We would be open to including the analysis above in the manuscript if it is deemed that it would be of value to do so.

Another area that the manuscript could be improved is expanding on how the complex might engage with chromatin and potentially providing some experimental evidence to support this. In Figure 6 they begin to cover this, but only a rough model is presented where some of the domains of specific proteins are highlighted that are known to have specific chromatin interactions. Here, the manuscript would greatly benefit from a more detailed discussion of the prior literature for these domains, for example. The authors should then discuss whether or not the specific regions, domains, and key amino acids mediating these interactions are accessible or buried in the structure that they present.

We presume the reviewer is referring to Figure 4 of the original manuscript. We would like to draw the reviewer's attention to the section titled "Flexibility and Positional Disorder in the Rpd3L Complex", where we already discussed these points in the original manuscript and referenced 5 studies describing the functions of the PAH1/2 domains of Sin3 as well as the PHD domains of Pho23 and Cti6 in engaging with sequence-specific DNA-binding repressors and H3K4me₃ signals, respectively. Since neither the PAH1/2 nor the PHD domains of the respective proteins are visible in our maps, we discussed the implications of these inherently dynamic associations in the revised manuscript. We also added new data showing that a conserved, basic region of Cti6 is involved in binding

to a H2A-H2B dimer that harbors the so-called acidic patch on nucleosomes (Supp. Fig. S8). We also discussed the implications of this finding in chromatin engagement by the Rpd3L complex.

Lastly, the manuscript would also benefit from an analysis of the experimental molecular weight of the complex they built the model on and compare it with the density map they obtained. This could be done in the supplement but Figure 1B looks to have a sizable amount of unassigned density and speculating on what might be in these areas is warranted and would also improve the manuscript. In figure 4 they appear to be fitting Ume1 into these areas, but a more detailed analysis and explanation would be helpful.

Based on known and determined stoichiometry of the Rpd3L complex, the complex would have an molecular weight of ~833kDa (without Ume6 and Ash1) and ~990kDa (with them). We were able to model ~312kDa from the high resolution 3.4A map and ~497kDa using the low-resolution map (where we included Ume1 and parts of Cti6). This information has been added to the legend to Supplementary Figure S1B.

Also, in the original manuscript we stated, "In silico modeling using AlphaFold2-multimer suggests that whereas the former segment comprising Dep1 and Sds3 likely serves as the binding interface for a helical segment at the C-terminus of Cti6, the latter most likely corresponds to a helical domain at the C-terminus of Sin3 that engages with the WD40 domain of Ume1 (Fig. 4A & Supp. Fig. S7)." The rationale for how we assigned Ume6 (and Cti6) to the flexible portions of lobes I and II can also be found in Supplementary Figures S4 and S7 as well as the model building section of the Methods section. We have also included the AF2 extended model found in Supplement Figure S7 as a Supplemental file.

Reviewer #3 (Remarks to the Author):

In this study, Patel et al. report the cryo-EM structure of the chromatin-associated yeast Rpd3L complex. Using TAP tag-mediated purification of the native yeast complex through the Rxt2 subunit, they isolate all canonical subunits of the complex and acquire and process cryo-EM data to a 3.5 Å resolution. Their analysis confirms previous studies reporting a two-lobe confirmation of the complex, each lobe containing the catalytic subunit Rpd3, as well as the Sin3 and Ume1 proteins. These results are also consistent with the previously described stoichiometry of the complex. Interestingly, they demonstrate that most components of the complex serve as scaffolding subunits, several of them exhibiting a flexible conformation, and extensive intra-complex surface contacts. Such conformation would make the stable complex amenable to recruitment by transcription factors and tethering to chromatin. The most important finding may be the discovery that one of the two Rpd3 subunits of the complex, namely the one associated with the Rxt3 lobe, is likely catalytically inactive as a result of steric hindrance from the Rxt2 subunit. Altogether, the results presented are clear and convincing. While the study mostly confirms previous models of the Rpd3L complex, it also provides novel insight on the respective contribution of each subunit of the complex, and by extension, of the mammalian homologs of these components.

Some concerns mitigate the enthusiasm for the study.

First, there is little attempt to explain known activities of specific subunits of the complex based on the structure elucidated here. For example, the authors do not speculate in depth on the molecular bases underlying the requirement for Sds3 in Rpd3L deacetylase activity. A more throughout discussion of the gain in knowledge obtained from the structure as it relates to the previously established properties of some of the subunits would be warranted.

We thank the reviewer for this suggestion. In the revised manuscript, we have addressed this concern by speculating on the molecular roles of the various subunits in the section titled “Comparison with Previous Genetic, Biochemical, and Structural studies”.

Second, a major part of the results obtained here is largely confirmatory of previous models of the Rpd3L complex structure. In particular, the study by Sardiù et al, PLoS ONE has suggested that the Rpd3L complex structure consisted of two asymmetric modules, each containing Rpd3 and Sin3 components.

We respectfully disagree that our studies are largely confirmatory of previous models of the Rpd3L complex, as to our knowledge, our cryo-EM structure is the first near atomic resolution structure of any fully assembled Class I HDAC complex described in the primary literature that has been subject to peer review; the structure therefore is by itself a hugely significant advance. We also refer the reviewer to our response to Reviewer 2 regarding previously proposed models. In the study by Sardiù et al., the authors do find that the Rpd3L complex contains two stoichiometric equivalents of Sin3, Rpd3 and Ume1 relative to the other components of the complex, consistent with our results. However, the authors in that study do not delve into this finding any further and we are unable to find any reference to a “structure consist[ing] of two asymmetric modules”.

Given the striking discovery that one of the two Rpd3 subunits may be inactive, a deeper understanding of this feature would greatly increase the impact of this study. It would thus be helpful for the authors to put forward and experimentally test a model that would justify the need for this inactive lobe in the Rpd3L canonical complex, or at least experimentally confirm that the presence of Rxt2 inactivates lobe2-Rpd3.

Our current hypothesis, which we did not discuss in the manuscript, is that the second (inactive) set of Sin3, Rpd3 and Ume1 in the Rpd3L complex either plays a role in nucleosome/chromatin engagement or Rpd3-II is allosterically activated by certain substrates binding to Rpd3-I. We note that Rxt2 appears to perform a critical scaffolding role for the assembly of the second lobe, so any mutagenesis experiment needs to be performed with considerable care. However, we have included new data that shows that our purified Rpd3L complex can deacetylate a model acetylated substrate and is responsive to known HDAC inhibitors (**Supp. Fig. 1C & 1D**). We completely agree with the reviewer that investigating this unique feature would be highly impactful. However, we also recognize that investigating the mechanisms of when and how the inhibition is relieved cannot be addressed in the short-term as we expect these studies to likely involve extensive experimentation and hence is beyond the scope of this manuscript.

REVIEWERS' COMMENTS

Reviewer #1 (Remarks to the Author):

The authors have corrected the two minor issues previously listed and furthermore provided adequate response in regards to the cryoem workflow.

As such, I have no further suggestions or concerns to be addressed. I maintain that, from my perspective, publication is recommended.

Reviewer #3 (Remarks to the Author):

The authors have addressed my previous concerns.